# Metagenomic/Metaproteomic Investigation of the Microbiota in Dongbei Suaicai, a Traditional Fermented Chinese Cabbage

**Yamin Zhang, Haiyang Yan, Xiangxiu Xu, Xiaowei Xiao, Yuan Yuan [ID], Na Guo, Tiehua Zhang, Mengyao Li, Ling Zhu, Changhui Zhao [ID], Zuozhao Wang and Haiqing Ye *[ID]**

College of Food Science and Engineering, Jilin University, 5333 Xi'an Road, Changchun 130062, China; 15075020624@163.com (Y.Z.); yanyanhyjlu@163.com (H.Y.); fanyang23@mails.jlu.edu.cn (X.X.); xiaoxw21@mails.jlu.edu.cn (X.X.); yuan_yuan@jlu.edu.cn (Y.Y.); guona@jlu.edu.cn (N.G.); zhangth@jlu.edu.cn (T.Z.); 15615785021@163.com (M.L.); zhulingzlwork@163.com (L.Z.); czhao@jlu.edu.cn (C.Z.); zuozhao@jlu.edu.cn (Z.W.)
* Correspondence: yehq@jlu.edu.cn

**Highlights:**

**What are the main findings?**

- Important metabolic pathways related to carbohydrate and amino acid metabolisms were enriched.
- Pediococcus was first found to be one of the most popular genera of late fermenta-tion.

**What is the implication of the main finding?**

- The multi-omics of metagenomics and metaproteomics was used to research the microbial ecosystem of Dongbei Suaicai.

**Abstract:** Dongbei Suaicai (DBSC) has a complicated microbial ecosystem in which the composition and metabolism of microbial communities during the process have not been well explored. Here, combined metagenomic and metaproteomic technology was used to reveal the taxonomic and metabolic profiles of DBSC. The results showed that firmicutes and proteobacteria were the prevalent bacteria in phylum and *Pseudomonas*, while *Weissella*, *Pediococcus*, and *Leuconostoc* were the prevalent genus. The vital metabolic pathways were involved in glycolysis/gluconeogenesis [path: ko00010], as well as pyruvate metabolism [path: ko00620], fructose and mannose metabolism [path: Ko00051], glycine, and serine and threonine metabolism [path: Ko00260]. Moreover, the key proteins (dps, fliC, tsf, fusA, atpD, metQ, pgi, tpiA, eno, alaS, bglA, tktA, gor, pdhD, aceE, and gnd) in related metabolized pathways were enriched during fermentation. This study will aid in facilitating the understanding of the fermentation mechanisms of DBSC.

**Keywords:** metagenomic; metaproteomic; Dongbei Suancai; fermentation; mechanism; omics

## 1. Introduction

Dongbei Suancai (DBSC), a traditional fermented Chinese cabbage (grown in large quantities all over China) prepared at low temperatures for 1–2 months, has been popular in China and neighboring countries such as South Korea, Japan, and Russia for several centuries [1]. The spontaneous fermentation of DBSC is a method of preservation of vegetables with a long-recorded history, which converts the Chinese cabbage to a product with a pleasant sour flavor [2]. Traditional DBSC is fermented based on the microorganisms attached to the raw material, which produces enzymes hydrolyzing carbohydrates, proteins, and lipids into organic acids and amino and acid aroma precursors [3,4]. Consequently, the fermentation of DBSC forms a complicated microbial ecosystem, which includes microorganisms, enzymes, and metabolites derived from substance and energy exchanges [5].



Previous studies have suggested that the diversity of microbial communities affects the physical and chemical properties of metabolites and enzyme compositions in DBSC [6]. However, up until now, specific mechanisms of substance metabolisms and functions of microbiota during the fermentation of DBSC have been unclear. Therefore, it is essential to deeply research DBSC for improving fermentation management.

With the development of molecular biology and genomics, metagenomics and metaproteomics are playing an important role in fermented product research, which has brought new opportunities for investigating complicated microbial communities [7,8]. Metagenomics refers to the study of genetic material of entire communities of organisms. This process usually involves third-generation sequencing (TGS) after the DNA extraction from the samples. TGS produces a large volume of data in the form of long reads to analyze the microbial community or predicted metabolic functions [9–11]. As a branch discipline studying the diversity of microbe flora and interactions within the environment, metagenomics has been applied to detected the dominant and low-epidemic genera to analyze the structural diversity and function of microbial communities, as well as the metabolic pathways related to specific environments in fermented food [12,13]. Meanwhile, metaproteomics is an emerging technology to study.

The metaproteomics technique is often focused on the quantitative functional makeup and associated microbes [14,15]. As a powerful means, metaproteomics has been applied to investigate the proteome in fermented fish, vegetable, maize products, and so on [8,16,17]. However, in metaproteomics, it is particularly challenging to find an appropriate database as the samples routinely contain a huge variety of microbial species, which results in a large size, redundancy, and a lack of biological annotations for the protein sequences [18]. A metagenomic-sequencing database from the same sample would increase the rate of protein identification. Therefore, in this work, the database is composed of protein amino acid sequences obtained by metagenomic analyses of the samples to be analyzed, which can improve the protein-identification rate [19]. Some studies have shown that multiomics can comprehensively unravel the mechanism of complicated microorganism communities [20,21].

In the present study, metagenomics and metaproteomics were applied together to analyze microbial communities and characterize the proteome with an aim to explore the microbial composition of DBSC metabolic activities occurring in the fermentation process and which enzymes perform these functions. This study will provide theoretical support for process regulation and quality DBSC product industries.

## 2. Materials and Methods

### 2.1. Sample Preparation and Collection

Fresh Chinese cabbages were purchased from the local market in Changchun, China and fermented using traditional methods [1]. First, the Chinese cabbages were cleaned by removing the rotten and dirty peels and subsequently air-drying them for 3 days, putting them into a 50 L fermentation jar, pickling them with 2% salt, placing a weight on the top, and spontaneously fermenting them (0–15 °C) for 60 days.

The samples ($n$ = 9) were collected at three fermentation time points (Days 20, 40, 60) and extracted from three replicate containers each time. At each sampling time, triplicate brine samples were drawn randomly from the top, middle, and bottom of one container and well mixed. A total of nine mixed samples were gathered and stored in sterile plastic bottles at −80 °C until testing [22].

### 2.2. Metagenomic Analysis

2.2.1. DNA Extraction, Library Construction, and Sequencing

Nine mixed brine samples were centrifuged at $13,400 \times g$ at 4 °C for 1 min, and then extracted for total DNA with Genomic DNA extraction kit (Tiangen–Bio Technology Co, Beijing, China, DP336). Then, 1 µg of DNA per sample was used as the input material for DNA sample preparation. After that, the DNA library with 300–500 bp insert size was constructed using the DNA library construction kits (Universal DNAseq Library Prep

Kit, Kaitai–Bio, Nanning, China). The libraries were sequenced on the Illumina HiSeq 2500 platform at the Kaitai Bioinformatics Technology Company (Thermo Fisher Scientific, Waltham, MA, USA).

### 2.2.2. Information-Analysis Process

Illumina raw reads were preprocessed using the FASTX–Toolkit software (Version 0.21.0) with the following treatments: (1) remove the splice sequences in the sequencing reads; (2) remove the reads containing more than 40% of unqualified (quality $\leq$ 15) nucleobase; (3) remove the sequences with a truncated length of less than 100 bp; (4) remove the sequences with the truncated N content of more than 5%. After quality filtering, clean reads were assembled and analyzed using the megahit software (Version 1.2.9). Afterwards, MetaGeneMark (http://topaz.gatech.edu/GeneMark/license_download.cgi accessed on 1 January 2023) was used to perform coding-region (CDS) predictions on individual samples and mixed-assembled contigs ($\geq$500 bp) with default parameters. Based on the CDS prediction results, non-redundant genes with identity $\geq$95% and coverage >90% were clustered, then the longest gene.final.count sequences were selected as representative sequences for the Unigene abundance statistics with Salmon software (Version 5.0.2). Using diamond software (Version 2.0.9) (blastp, value $\leq 1 \times 10^{-5}$) to compare the Unigenes with the non-redundant (NR in NCBI) database, the comparison results (evalue $\leq$ minimum evalue-10) were selected for species classification. Clusters of orthologous groups (COG) and Kyoto Encyclopedia of Genes and Genomes (KEGG) were used for functional analysis (Figure 1).

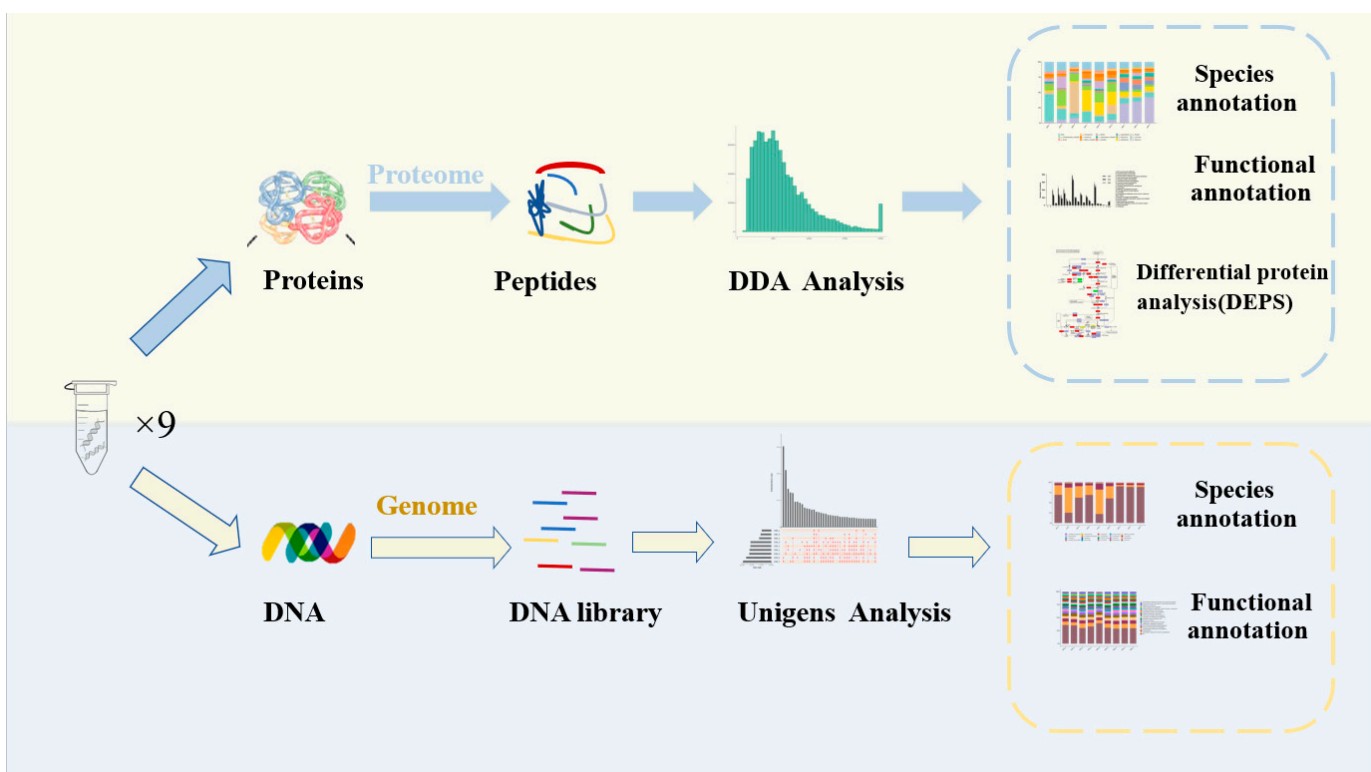

**Figure 1.** Schematic overview of the application of metagenomics and metaproteomics for the study of the DBSC.

### 2.3. Protein Preparation and Metaproteomics

2.3.1. Protein Extraction and Peptide Digestion

Proteins were extracted from each sample; an appropriate amount of SDT (4% SDS, 100 mM Tris-HCl, pH 7.6) lysate was added and quantified with Bicinchoninic Acid Assay (BCA) method. The 20 μg proteins were taken from each sample, added to 5X buffer and boiling bath water for 5 min, then separated by SDS-PAGE (4–20% precast gradient

gel, constant pressure 180 V, 45 min) and stained with Cauloblue R-250. The appropriate amount of protein was extracted from all samples and mixed into pool samples for construction of spectral library. All sample proteins were trypsin digested using the filter-aided proteome preparation (FASP). The peptides from the pool samples were graded with high-PH reversed-phase Peptide Separation Kit (10 levels) (Thermo Fisher Scientific, Waltham, MA, USA), desalted with C18 Cartridge, redissolved with 40 µL of 0.1% formic acid after lyophilized, and finally determined at OD280.

### 2.3.2. Mass Spectrometric Methods

The peptides were chromatographed by using a nanoliter flow-rate HPLC system (Easy-nLC 1200). Specifically, samples were injected into a C18 column for linear gradient separation [Thermo Scientific, ES802, 1.9 µm, 75 µm × 20 cm, 0.1% formic acid acetonitrile aqueous solution] [acetonitrile is 84%], 300 nL/min flow rate). Then, Q-Exactive HF-X mass spectrometer (Thermo Fisher Scientific, Waltham, MA USA) was used for mass spectrometry (detection mode: positive ion; primary mass spectral scan range: 350–1800 $m/z$; mass spectral resolution: 60,000 [@$m/z$ 200]; AGC target: $1 \times 10^6$, maximum IT: 50ms, dynamic exclusion time: 10 s). Each full MS scan collects 20 ddMS2 scans based on the inclusion list (isolation window:1.5 $m/z$; MS resolution: 30,000 [@$m/z$ 200]; AGC target: $1 \times 10^5$; maximum IT: 50 ms; MS2 activation type: HCD; normalized collision energy: 30 eV).

### 2.3.3. MS Data Analysis

DDA data were directly imported into the Spectronaut software (SpectronautTM 14.4.200727.47784). Then, the database was downloaded according to the corresponding species. The search parameters were set as follows: trypsin as enzyme, max miss cleavage site as 1, fixed modification as carbamidomethyl (C), dynamic modification as oxidation (M) and acetyl (protein N-term). In addition, the identified proteins needed to pass the set-filtering parameter FDR < 1%.

### *2.4. Bioinformatics Analysis*

The COG database (http://www.ncbi.nlm.nih.gov/COG accessed on 1 January 2023), KEGG database (http://www.genome.jp/kegg/pathway.html accessed on 1 January 2023), and NCBI-nr database (ftp://ftp.ncbi.nlm.nih.gov/blast/db/FASTA/nr.gz accessed on 1 January 2023) were used for bioinformatic analysis.

### 3. Results

### *3.1. Overview of Metagenomic Data*

The nine samples of DBSC were analyzed using metagenomic sequencing. This process generated 699,857,802 raw reads, of which, 683,407,618 reads remained after quality control. In total, >95% reads had sequencing errors lower than 1%, and 1,006,177 ORFs were acquired after gene prediction Table 1, which shows the high quality of the sequencing procedure and statistics of the reads.

**Table 1.** Statistics of sequencing and bioinformatics analysis.

| Eigenvalue | Data |
|---|---|
| **Sequencing** | |
| Raw reads(num) | 699,857,802 |
| Raw bases(bp) | 104,978,670,300 |
| Q20(%) | 96.77 |
| Q30(%) | 91.24 |
| Filter reads(num) | 683,407,618 |
| Filter bases(bp) | 101,969,988,159 |
| Q20(%) | 97.09 |
| Q30(%) | 91.64 |
| Valid% | 97.18 |
| **Assembly** | |
| Contigs (>=1000 bp) | 77,054 |
| Contigs (>=5000 bp) | 2302 |
| Contigs (>=10,000 bp) | 651 |
| Contigs (>=25,000 bp) | 158 |
| Contigs (>=50,000 bp) | 35 |
| Total length (>=25,000 bp) | 6,798,282 |
| Total length (>=50,000 bp) | 2,736,588 |
| Contigs | 241,723 |
| Largest contig | 163,143 |
| Total length | 262,961,360 |
| GC (%) | 52.75 |

**Table 1.** *Cont.*

| Eigenvalue | Data |
|---|---|
| N50 | 1143 |
| N75 | 736 |
| L50 | 59,782 |
| L75 | 132,629 |
| **Annotation** | |
| ORFs NO. | 1,006,177 |
| Integrity start | 165,987 (16.50%) |
| Integrity end | 307,836 (30.59%) |
| Integrity all | 445,458 (44.27%) |
| Integrity none | 86,896 (8.64%) |
| Total Length (bp) | 649.6 |
| Average Length (bp) | 645.61 |
| GC percent | 54.96% |

*3.2. Taxonomic Composition of Predicted Genes*

Figure 2A shows the existence of the top-20 phylum throughout the fermentation period. Firmicute was the most abundant phylum (61.13%), followed by proteobacteria (29.34%), uroviricota (0.39%), bacteroidetes (0.06%), actinobacteria (0.06%), ascomycota (0.02%), cyanobacteria (0.01%), and others. Meanwhile, the abundance of firmicute increased from 52.31% (Day 20) to 89.24% (Day 60) following fermentation progress. Conversely, the abundance of proteobacteria decreased from 37.77% (Day 20) to 4.14% (Day 60). At the family level, leuconostocaceae was expressed as the dominant family with an abundance of 43.15% at the 20th day of fermentation. However, with the extension of fermentation time, the abundance of leuconostocaceae obviously decreased to 28.10% on the 40th day and to 4.02% on the 60th day. Similar to leuconostocaceae, yersiniaceae was also a dominant family on the 20th day with an abundance of 10.85%; it decreased to 7% on the 40th day and to 0.93% on the 60th day. During the first 40 days of fermentation, the abundance of pseudomonadaceae increased from 14.60% to 18.77%, which played an important role in the early stage of fermentation, while it decreased to 1.13% on the 60th

day. It is noteworthy that lactobacillaceae gradually occupied the dominant role in the fermentation process, which accounted for an abundance from 2.96% (Day 20) to 18.37% (Day 40), and up to 81.80% at Day 60, which played a decisive role in the late fermentation of DBSC (Figure 2B).

At the genus level (Figure 2C), the abundance of *Pediococcus* (belonging to lLactobacillaceae) remarkably increased up to 67.19% on the 60th day from 0.04% (Day 20) and 0.19% (Day 40), and it became the most popular genera in the final period. As a member of the lactobacillaceae, *Latilactobacillus* increased up to 15.15% (Day 40) from 1.66% (Day 20), then decreased to 2.93% (Day 60). As important members of leuconostoceae, *Leuconostoc* and *Weissella* declined with the prolongation of fermentation from 28.39% (Day 20), to 16.75% (Day40), to 3.76% (Day 60), and from 14.68% (Day 20), to 10.93% (Day 40), to 0.17% (Day 60), respectively. *Pseudomonas* shares the same trend as pseudomonadaceae, which increases from 14.60% (Day 20) to 18.77% (Day 40), yet it decreased to 1.13% on the 60th day. As shown in Figure 2D, *Weissella_soli* and *Leuconostoc_mesenteroides* were the main species on the 20th day, and their abundance was 16.59% and 9.92%, respectively. Similar to *Weissella* and *Leuconostoc*, the abundance of *Weissella_soli* decreased to 10.28% on Day 40 and abruptly decreased to 0.13% on Day 60, and the abundance of *Leuconostoc_mesenteroides* decreased to 5.92% on Day 40 and 1.34% on Day 60. *Lactobacillus_curvatus* occupied the position of the most dominant species on the 40th day with the abundance of 12.92%, then it dropped to 2.39% on Day 60. *Pediococcus_parvulus* was almost absent during the first 40 days' fermentation, with abundances of 0% (Day 20) and 0.10% (Day 40), respective. However, it abruptly increased to 54.81% on the 60th day, occupying over half of all species.

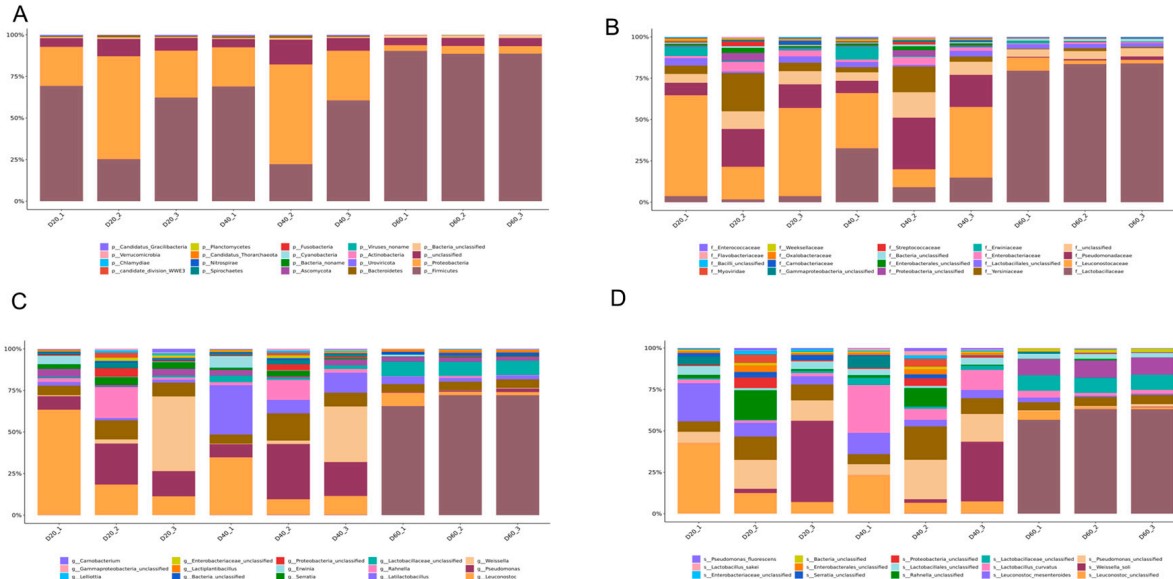

**Figure 2.** Genomics-based taxonomic profiles: relative abundance (percentage) of microbes at the phylum (**A**), family (**B**), genus (**C**), and species (**D**) level during DBSC fermentation. The abscissa (20, 40, 60) represented fermentation time (day). Metagenomic sequences were compared against the NCBI-nr database. Overall, 92 phyla, 561 families, 1658 genera, and 7198 species were recovered and identified (Supplementary Material A). The top-20 relative abundance of phylum, family, genera, and species that dominated at each fermentation time point are shown in Figure 2.

### 3.3. Function Classification of Identified Genes

The abscissa (20, 40, 60) represented fermentation time (day). Figure 3 (A) shows the 23 functional categories of identified genes that were annotated by the COG database. The top-five categories were nothing analysis (NA), translation, ribosomal structure, and biogenesis (J), transcription (K), carbohydrate transport and metabolism (G), and amino acid transport and metabolism (E). Obviously, the percentage of most categories

remained steady during fermentation. Among them, the "carbohydrate transport and metabolism" category accounted for 5% on Day 20, 5.01% on Day 40, and increased to 6.27% on Day 60. The "amino acid transport and metabolism" category accounted for 5.50%, 5.06%, and 4.54% on Days 20, 40, and 60. To generate deeper insights into the functional pathways associated with identified genes, they were mapped to the KEGG pathways of KEGG database. The top-20 metabolic subsystems at the KEGG classification Level 3 were identified within the annotated dataset (B). These pathways mainly belong to four classification of 1 level, as follows environmental information processing: ABC transporters (path02010), two-component system (path02020), phosphotransferase system (PTS) (path02060); genetic information processing: ribosome (path03010), aminoacyl-tRNA biosynthesis (path00970); cellular processes: quorum sensing (path02024); metabolism: purine/metabolism (path00230), pyrimidine metabolism (path00240), starch and sucrose metabolism (path00500), amino sugar and nucleotide sugar metabolism (path00520), fructose and mannose metabolism (path00051), galactose metabolism (path00052), glycolysis/gluconeogenesis (path00010), pyruvate metabolism (path00620), cysteine and methionine metabolism (path00270), glycine, serine and threonine metabolism (path00260), peptidoglycan biosynthesis (path00550), oxidative phosphorylation (path00190), and fatty acid biosynthesis (path00061) (Supplementary Material B).

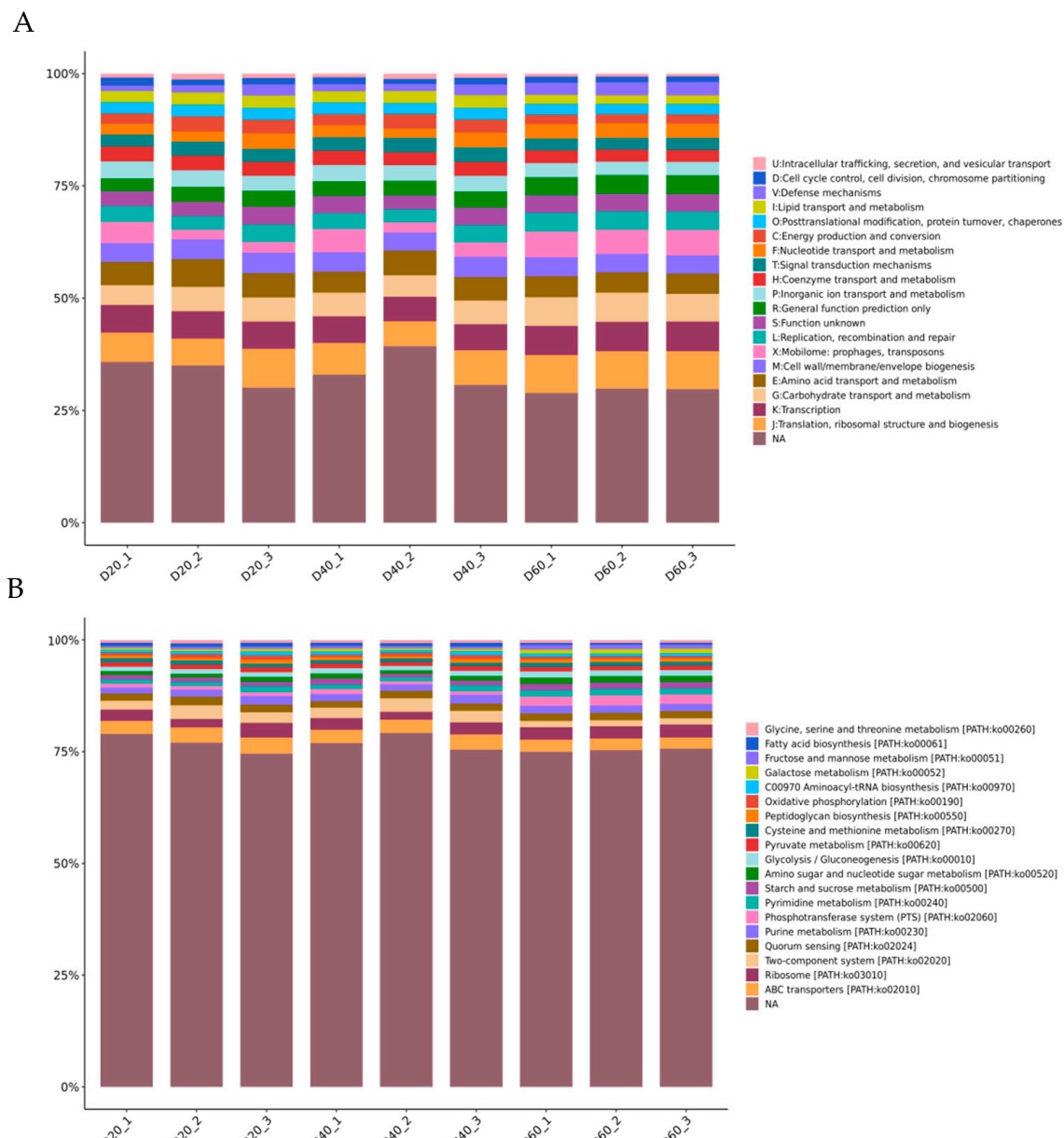

**Figure 3.** Function classification of predicted genes: relative abundance (percentage) of function classification COG (**A**) and KEGG (**B**) during DBSC fermentation.

### 3.4. Taxonomic Composition of Identified Proteins

Through DDA quantitative proteomics technology, 9944 proteins with a high confidence of the correct peptide-sequence assignment (false discovery rate [FDR] < 0.01) were examined totally, including 7719 on Day 20, 6400 on Day 40, and 6132 on Day 60, respectively. The abundances of phyla and genera in the analyzed samples proteins are shown in Figure 4. The results showed that 19 phyla were detected from the nine samples. Among these phyla, firmicutes (89.58%) and proteobacteria (9.63%) accounted for 99.21%. Following the most predominant phyla, the others were ascomycota (0.35%), bacteroidetes (0.20%), and non-dominant phyla (less than 0.01%) (Figure 4A). At the genus level, Figure 4B shows that the proteins were mainly derived from *Weissella* (27.00%), *Leuconostoc* (19.81%), *Pseudomonas* (14.58%), *Rahnella* (6.58%), and *Pediococcus* (5.97%) on the 20th day. However, on the 40th day of fermentation, *Latilactobacillus* (25.07%) occupied the position of the most abundant genus, followed by *Pseudomonas* (12.48%), *Leuconostoc* (11.51%), *Weissella* (6.70%), *Rahnella* (6.04%), and so on. It is worth noting that *Pediococcus* (36.28%) was found to be the

most abundant genus, along with a number of major genera of *Loigolactobacillus* (9.69%), *Leuconostoc* (7.60%), *Latilactobacillus* (7.29%), *Lactobacillaceae_unclassified* (5.89%), and others, which had an abundance of <5% on the 60th day (Supplementary Material C).

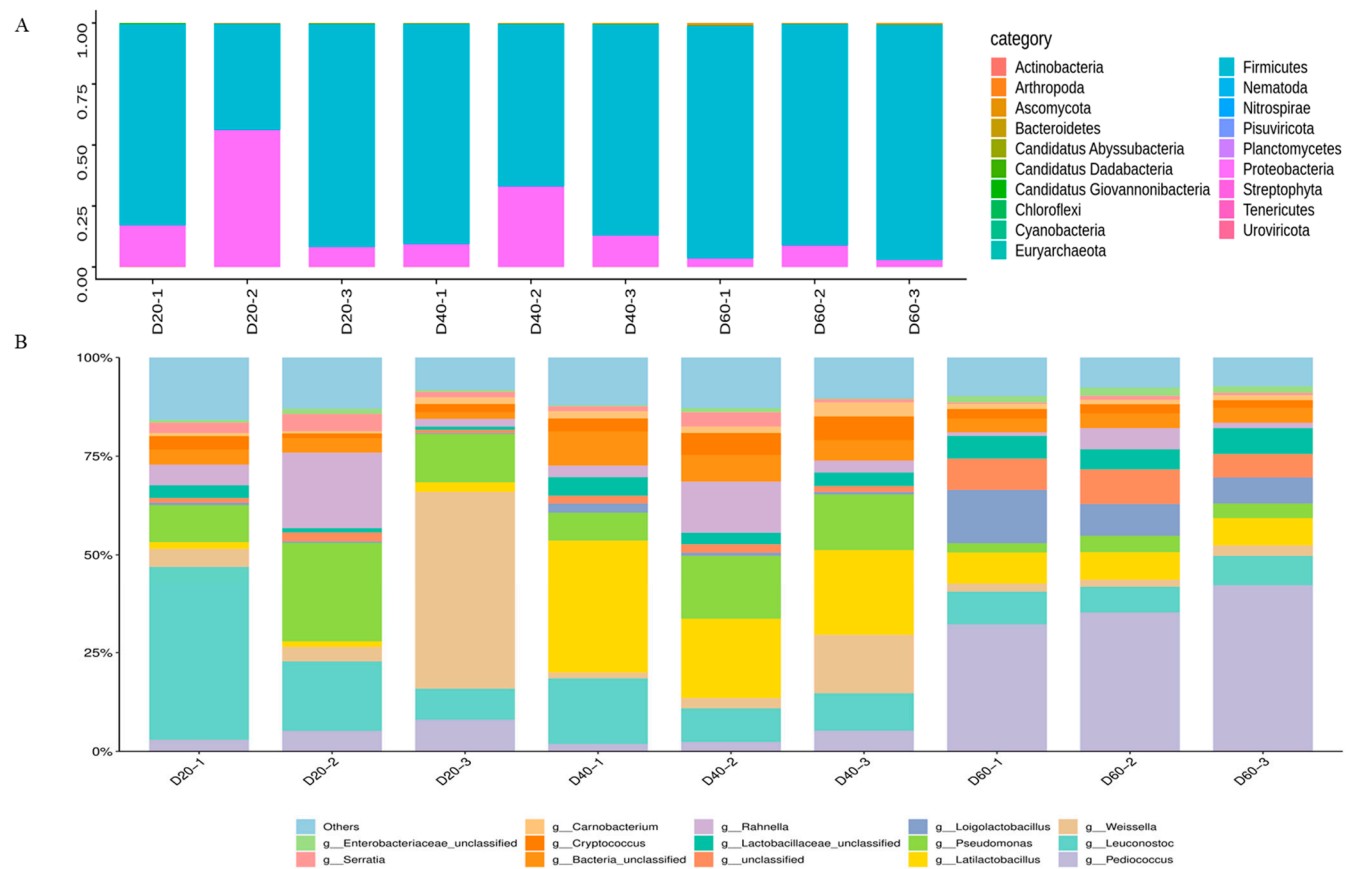

**Figure 4.** Protein-based taxonomic profiles: relative abundance (percentage) of microbial at the phylum (**A**) and genus (**B**) level during DBSC fermentation. The abscissa (20, 40, 60) represented fermentation time (day).

### 3.5. Function Classification of Proteins

According to the COG database, the identified proteins were functionally clustered into 22 classifications (Figure 5A). Among them, the "translation, ribosomal structure, and biogenesis" category was assigned the most proteins, which were 2311 (17.36%), 1987 (18.74%), and 1898(18.67%) om Days 20, 40, and 60, respectively. They were followed by 1206 (9.06%, Day 20) 895 (8.44%, Day 40), and 907 (8.92%, Day 60) proteins, which were enriched in the "transport and metabolism" category. Furthermore, the "amino acid transport and metabolism" category was related to proteins that decreased from 1318 (9.90%, Day 20) to 804 (7.58%, Day 40) to 634 (6.24%, Day 60). The other categories accounted for a relatively small proportion. The function annotation of identified proteins in nine DBSC samples was performed using KEGG. Top-10 KOs (K03286, K04077, K02406, K03671, K02358, K04043, K03704, K02112, K02078, K02935) were shown in Figure 5B. By analyzing the homologous classification of proteins, the 10 proteins identified were involved in genetic information and environmental treatment pathways, which necessitates further exploration (Supplementary Material D).

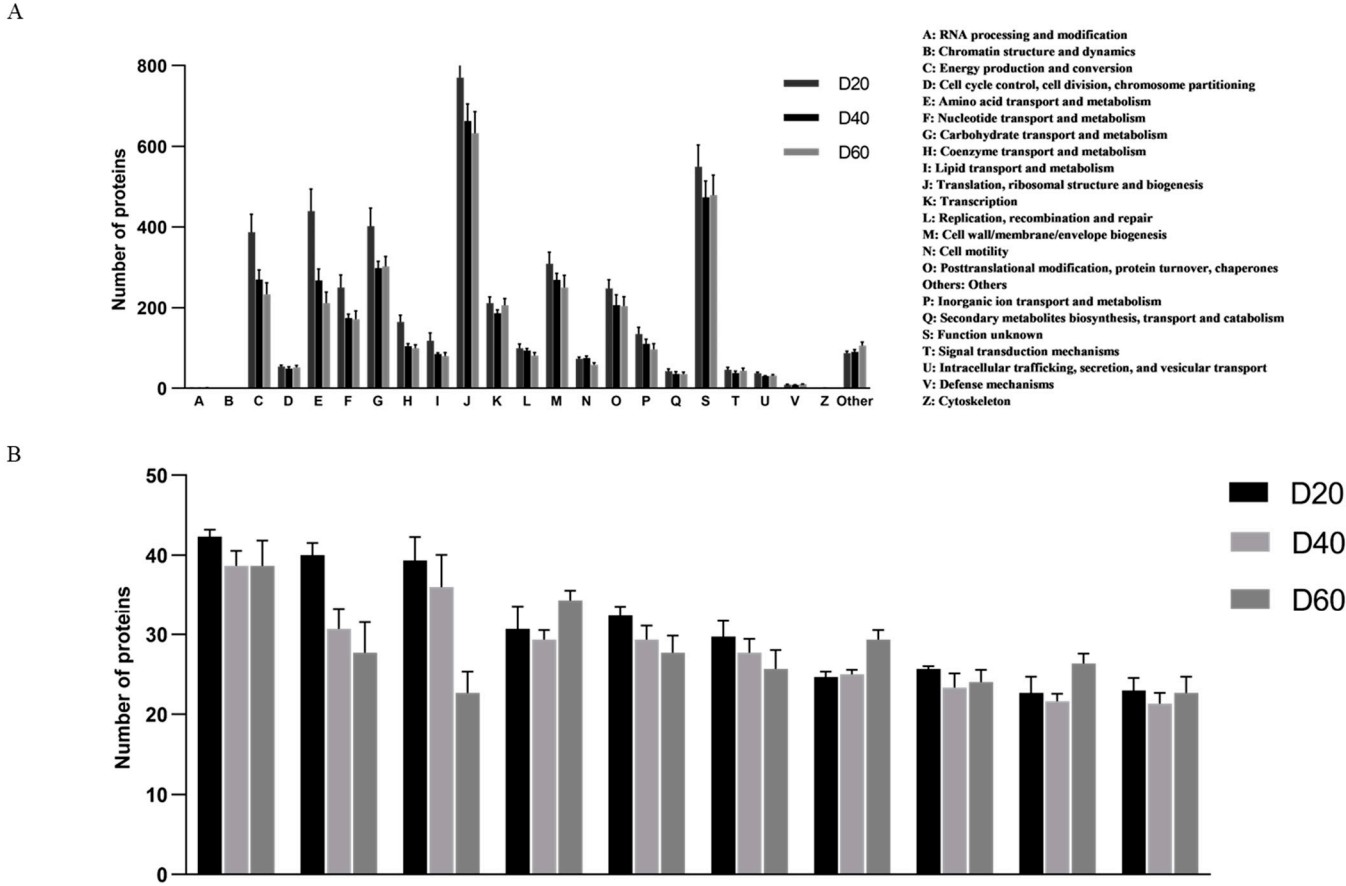

**Figure 5.** Function classification of proteins: relative abundance (percentage) of function classification COG (**A**) and Top-10 KEGG KO (**B**) during DBSC fermentation. Different colors represent different fermentation times (20, 40, 60). Remark: (1) TC.OOP; OmpA−OmpF porin, OOP family (K03286), (2) groEL, HSPD1; chaperonin GroEL (K04077), (3) fliC; flagellin (K02406), (4) trxA; thioredoxin 1 (K03671), (5) tuf, TUFM; elongation factor Tu (K02358), (6) dnaK, HSPA9; molecular chaperone DnaK(K04043), (7) cspA; cold-shock protein (beta−ribbon, CspA family (K03704), (8) TPF1B, atpD; F−type H+−transporting ATPase subunit beta [EC:3.6.3.14] (K02112), (9) acpP; acyl carrier protein (K02078), (10) RP−L7, MRPL12, rplL; large subunit ribosomal protein L7/L12 (K02935).

### 3.6. Overview and Analysis of Differential Proteins

Based on the standard (fold change >2 or <0.5 and $p < 0.05$ ($t$-test)), the numbers of significantly upregulated and downregulated proteins between different samples are shown in Figure 6A, of which by more than 10 times was marked with darker colors. The upregulated and downregulated DEPs were 150 and 184 in D20 versus D40, 109 and 317 in D20 versus D60, 85 and 96 in D40 versus 60, respectively. Furthermore, the top-10 DEPs of the three comparison groups were shown in Figure 6B–D. To engender insights into the functional and species information of these DEPs, the DEPs were classified to the KEGG and NR database (Table 2). Primary functional categories of the DEPs were associated mainly with carbohydrate metabolism, amino acid metabolism, signal transduction, and transcription and translation pathways. Specifically, the proteins (pgi, tpiA, bglA, gor, pdhD, aceE, gnd,) that were involved in the pathways of starch and sucrose metabolism (map00500), glycolysis/gluconeogenesis (map00010), galactose metabolism (map00051), pentose phosphate pathway (map00030), and so on, were mainly produced by *Pediococcus*, *Pseudomonas*, and *Lactococcus*. In addition, the proteins (SOD$_2$, fusA, atpD, oppA, rbsB, groEL, Dps, fliC, tsf, metQ) related to signal transduction, transcription, and translation pathways were mainly secreted by *Pediococcus*, *Pseudomonas*, *Weissella*, *Rahnella*, and *Serratia*.

Only pdhD, mainly produced by *Lactobacillus* and *Carnobacterium,* was mapped to glycine, serine, and threonine metabolism (map00260) pathways (Supplementary Material E).

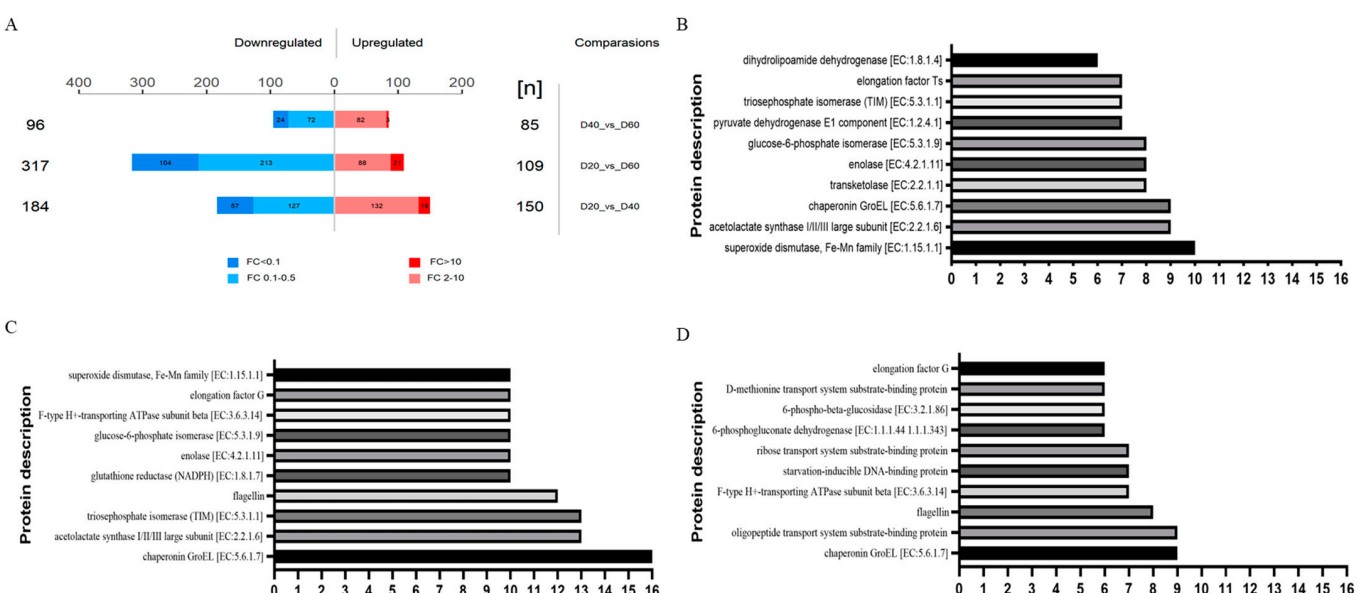

**Figure 6.** Differential proteins (DEP) analysis: the differential of protein number between three time points (**A**), KEGG classification statistics of D20 VS D40 (**B**), D20 VSd60 (**C**), and D40 VS d60 (**D**) differen-tial proteins. Remark: oligopeptide transport system substrate-binding protein (oppA, mppA); ribose transport system substrate-binding protein (rbsB); superoxide dismutase, Fe–Mn family [EC: 1.15.1.1] (SOD2); chaperonin GroEL [EC: 5.6.1.7] (groEL, HSPD1); starvation-inducible, DNA-binding protein (dps); flagellin (fliC); Elongation Factor Ts (tsf); Elongation Factor G (fusA); F-type H+-transporting ATPase subunit beta [EC: 3.6.3.14] (atpD); D-methionine transport system substrate-binding protein (metQ); glucose-6-phosphate isomerase [EC: 5.3.1.9] (pgi); triosephosphate isomerase (TIM) [EC: 5.3.1.1] (tpiA); enolase [EC: 4.2.1.11] (eno); acetolactate synthase I/II/III large subunit [EC: 2.2.1.6] (ilvB, ilvG, ilvI); 6-phosphovsbetavsglucosidase [EC: 3.2.1.86] (bglA); transketolase [EC: 2.2.1.1] (tktA, tktB); glutathione reductase (NADPH) [EC: 1.8.1.7] (gor); dihydrolipoamide dehydrogenase [EC: 1.8.1.4] (pdhD); pyruvate dehydrogenase E1 component [EC: 1.2.4.1] (aceE); 6-phosphogluconate dehydrogenase [EC: 1.1.1.44, 1.1.1.343] (gnd).

**Table 2.** DEP metabolic pathways and species-statistics analysis.

| DEP | DEPperid | KO | KEGG PATHWAY | Main Species |
|---|---|---|---|---|
| oppA | D40vsD60 | K15580 | map01501, map02010, map02024 | *Pediococcus* |
| rbsB | D40vsD60 | K10439 | map02010, map02030 | *Pseudomonas* |
| SOD₂ | D20vsD40 D20vsD60 | K04564 | map04013, map04068, map04146, map04211, map04212, map04213, map05016 | |
| groEL | D20vsD40 D40vsD60 D20vsD60 | K04077 | map03018, map04212, map04940, map05134, map05152 | *Xanthomonas* |
| dps | D40vsD60 | | | *Serratia* |
| fliC | D40vsD60 D20vsD60 | K02406 | map02020, map02040, map04621, map04626, map05132, map05134 | *Pseudomonas* |
| tsf | D20vsD40 | K02357 | map03018 | *Pseudomonas* |
| fusA | D40vsD60 D20vsD60 | K02112 | | *Pseudomonas* |

**Table 2.** *Cont.*

| DEP | DEPperid | KO | KEGG PATHWAY | Main Species |
|---|---|---|---|---|
| atpD | D40vsD60 D20vsD60 | K02112 | map00190, map00195, map01100 | *Pseudomonas* |
| metQ | D40vsD60 | K02073 | map02010 | *Weissella Rahnella* |
| pgi | D20vsD40 D20vsD60 | K01810 | map00010, map00030, map00500, map00520, map01100, map01110, map01120, map01130, map01200 | *Serratia Leuconostoc* |
| tpiA | D20vsD40 D20vsD60 | K01803 | map00010, map00051, map00562, map00710, map01100, map01110, map01120, map01130, map01200, map01230 | *Serratia Lactococcus* |
| eno | D20vsD40 D20vsD60 | K01689 | map00010, map00680, map01100, map01110, map01120, map01130, map01200, map01230, map03018, map04066 | *Pseudomonas Leuconostoc* |
| alaS | D20vsD40 D20vsD60 | K01652 | map00290, map00650, map00660, map00770, map01100, map01110, map01130, map01210, map01230 | *Leuconostoc Rahnella* |
| bglA | D40vsD60 | K01223 | map00010, map00500 | *Pediococcus* |
| tktA | D20 vs D40 | K00615 | map00030, map00710, map01051, map01100, map01110, map01120, map01130, map01200, map01230 | *Serratia Pseudomonas* |
| gor | D20 vs D60 | K00383 | map00480, map04918 | *Pseudomonas Serratia* |
| pdhD | D20vsD40 | K00382 | map00010, map00020, map00260, map00280, map00620, map00630, map00640, map01100, map01110, map01120, map01130, map01200 | *Lactobacillus Carnobacterium* |
| aceE | D20vsD40 | K00163 | map00010, map00020, map00620, map01100, map01110, map01120, map01130, map01200 | *Pseudomonas* |
| gnd | D40vsD60 | K00033 | map00030, map00480, map01100, map01110, map01120, map01130, map01200 | *Pseudomonas* |

## 4. Discussion

### 4.1. Microbial Community Composition

Spontaneous fermentations generally lead to distinctive, complicated, and well-balanced microbial communities [4]. Our results showed that protebacteria and firmicutes were the dominant phyla, which was consistent with previous reports [23,24]. Based on the metagenomic date, during DBSC spontaneous fermentation, there was a dramatic decrease in the phylum proteobacteria (from 52.31% to 89.24%) and an increase of firmicutes (from 52.31% to 89.24%). The microbial community structure mainly relies on the fermentation environment [25]. A decrease of pH can lead to the microbial community changing to enrich those bacteria with more acidic tolerance during fermentation [26]. In DBSC fermentation,

lactobacillaceae are considered as efficient fermenters, which are proficient in metalizing carbohydrates to generate a range of organic acids under anaerobic or limited oxygen conditions [27]. Moreover, lactobacillaceae have the ability to survive in self-imposed acid stress, which explains how lactobacillaceae abounded in the late stage of fermentation [28].

Furthermore, *Leuconostoc* spp. is a gram-positive, non-sporulating, non-gelatin liquifying and microaerophilic lactic acid bacteria, which is also considered as a strong food fermentation starter [29]. *Leuconostoc* can ferment glucose into lactic acid, carbon dioxide, and acetic acid and ethanol [30]. Zabat et al. showed that the initial proliferation of *Leuconostoc* rapidly produced carbon dioxide and acid, which quickly lowered the environmental pH, inhibiting the growth of undesirable microorganisms that might cause food spoilage while preserving the color of the cabbage, as well as favoring the succession of other LABs [31]. This agrees with our results that *Leuconostoc* was identified as the main species on Day 20 of DBSC fermentation. As a species of *Leuconostoc*, *Leuconotoc mesenteroides* can ferment fructose into mannitol, which are frequently used as sweeteners and thickeners in food sectors and is tolerant during osmotic stress [32]. And *Leuconotoc mesenteroides* can also utilize other sugars (pentose, arabinose, and xylose), which contributes to the development of complex and unique flavor characteristics [30].

It is worth noting that *Pediococcus* suddenly turned into the dominant genera in the later period of DBSC fermentation. This result was inconsistent with other reports that *Lactobacillus* was generally considered the predominant bacterial genus, while *Pediococcus* plays an unimportant role in DBSC samples [23,33]. Tlais et al. have suggested that lactobacilli and *Pediococcus* all have high metabolic potential for phenolics bioconversion in DBSC fermentation [34]. Additionally, *Pediococcus* could consume glucose and fructose as the main fermentable sugars to produce lactic acid and acetic acid, which compose the main organic acid of DBSC [35]. Therefore, we suggest that *Pediococcus* and *Lactobacillus* may have similar functions in the fermentation process. Interestingly, as the member of *Pediococcus*, *P. parvulus* has been reported as a novel key player during Xuanwei Ham fermentation, including the secretion of acid, expedition of the growth of LAB, and the consumption of sugar [36]. This indicates that the function of the *Pediococcus* genus has always been neglected, which may have an important influence on the flavor formation in the later period of DBSC fermentation.

According to the metaproteomic results, at the level of phylum, ascomycota was the only fungi among the dominant phylum, which may be due to ascomycota fungi adapting well to saprobic, parasitic, or mutualistic life modes [37]. However, it is a pity that prior studies have focused primarily on bacterium that limited knowledge of the fungi role in the fermentation of fermented vegetables, which should be paid more attention in the future [38].

*Pseudomonas* is a common genus in food, which can be isolated from plant surfaces, soil, water, and raw materials and may be frequently introduced into the processing environment through many routes [39]. *Pseudomonas* form biofilms at low (4 and 10 °C) temperatures, which is a survival mechanism that can allow bacteria to withstand the external stresses, but *Pseudomonas* growth is limited in acidic and high-salt environments [40]. Previous studies have considered that *Pseudomonas* is the important genus for the spoilage of food, which can produce large amounts of extracellular enzymes and degraded foods, resulting in off-flavors and off-tasting foods [41,42]. *Pseudomonas* has a strong ability of producing proteases and possesses genes for carbohydrate transport and metabolism, signal transduction, secondary metabolite synthesis, polysaccharide lyase, and carbohydrate esterases [43,44]. In the study, a high abundance of *Pseudomonas* was observed during the first 40 days, while the abundance gradually decreased due to the anaerobic acidic environment.

*Weissella* are facultatively anaerobic chemoorganotrophs with an obligately fermentative metabolism that has characteristics for fermenting glucose heterofermentative via the hexose-monophosphate and phosphoketolase pathways that can produce lactic acid [45]. In our work, *Weissella* were initially assigned to the most proteins, which consisted with reports from Yang et al. [26]. *Weissella* have been isolated from and occur in a wide range

of habitats, such as European sourdoughs and African traditional fermented foods. So, *Weissella* play a perceived technical role in such traditional food fermentation.

Metagenomic and metaproteomic investigations of main microbial had no significant divergence, which included *Pseudomonas, Weissella, Pediococcus,* and *Leuconostoc.* Meanwhile, the content of the same phylum or genus was different in the two quantification strategies. It is reasonable that taxonomic abundances based on metagenomics and metaproteomics are different since there are divergences between genetic potential and expressed functional activity [46]. In some cases, the proportion of gene species is not equal to functional microorganisms [47]. The reliability of the results was greatly improved by combining two methods to verify each other.

*4.2. Biological Function Analysis*

Remarks: The box with a red background denotes an upregulated protein, the box with a green background denotes downregulated proteins, and the box with a yellow background denotes the presence of numerous proteins that are both upregulated and downregulated. Large squares stand in for other pathways, while light-green bottom boxes represent species-specific proteins and light-purple bottom boxes indicate small molecular metabolites.

Functional quantification is increasingly recognized as the essential link between biodiversity patterns and ecosystem functioning [48]. Overall, during DBSC fermentation, a large amount of proteins (enzymes) were involved in life activities, such as DNA replication, RNA transcription and protein synthesis, cellular activities, intracellular and intercellular communication, energy metabolism, and activities related to the metabolism of substances (carbohydrates, amino acids). In particular, proteins (fusA, atpD, oppA, rbsB groEL, Dps, fliC, tsf, metQ) related to basic life activities were the most abundant and produced by a variety of bacteria, such as *Pseudomonas, Weissella, Pediococcus,* and *Leuconostoc.* These results suggested that microorganisms constantly metabolize and reproduce, communicate with the surrounding environment, and produce stress reactions in the DBSC fermentation process. In addition, four co-occurrence-enriched metabolic pathways were explored, including glycolysis/gluconeogenesis [PATH: ko00010], pyruvate metabolism [PATH: ko00620], fructose and mannose metabolism [path: Ko00051], glycine, and serine and threonine metabolism [path: Ko00260].

The glycolysis/gluconeogenesis pathway is the process of converting glucose into pyruvate and generating small amounts of ATP and NADH, and it is also a central pathway that produces important precursor metabolites [49]. As shown in Figure 7A, the glycolysis/gluconeogenesis pathway includes glycolysis (glucose => pyruvate), gluconeogenesis (oxaloacetate => fructose-6P), pyruvate oxidation (pyruvate => acetyl-CoA), and the pathway modules. Additionally, the enzymes (glucose-6-phosphate isomerase [EC:5.3.1.9], triosephosphate isomerase [TIM] [EC:5.3.1.1], 6-phospho-beta-glucosidase [EC:3.2.1.86], dihydrolipoyl dehydrogenase [EC:1.8.1.4], and pyruvate dehydrogenase E1 component [EC:1.2.4.1]) involved in glycolysis/gluconeogenesis pathways are presented in Figure 7A (marked with red circles), which mostly belonged to *Leuconostoc, Serratia,* and *Pediococcus.* Generally speaking, the microorganisms present in the raw material undergo homofermentation or heterofermentation, with the formation of various products in DBSC. Sauerkraut undergoes a sequential fermentation that is initiated by heterofermentative lactic acid bacteria and completed by homofermentative bacteria [50]. Our results showed that the glycolysis/gluconeogenesis pathway had similar metabolic levels at Days 20, 40, and 60 during DBSC fermentation, which was likely due to the fact that *Leuconostoc* dominated heterofermentation in the first 20 days, and *Serratia* played a vital role in metaphase, as well as *Pediococcus*-dominated homofermentation in the later stage [51,52]. This result indicated that the three genera contributed to carbohydrate degradation to generate flavor in DBSC. Pyruvate metabolism (pathway 00620) plays an important pivotal role in the metabolic linkage under the presence of dihydrolipoamide dehydrogenase [EC: 1.8.1.4] and pyruvate dehydrogenase [EC: 1.2.4.1] (Figure 7B). Pyruvate metabolism is usually an important intermediate process in the carbohydrate metabolic pathways and the formation

of various amino carbon skeletons, mainly from the carbohydrate metabolism [53]. In DBSC fermentation, pyruvate was converted primarily to lactic acid [54]. Although lactic acids are odorless and do not directly contribute to the flavor of fermented products, they are responsible for cider–vinegar flavor, ethyl lactate, ethyl acetate, and so on [50].

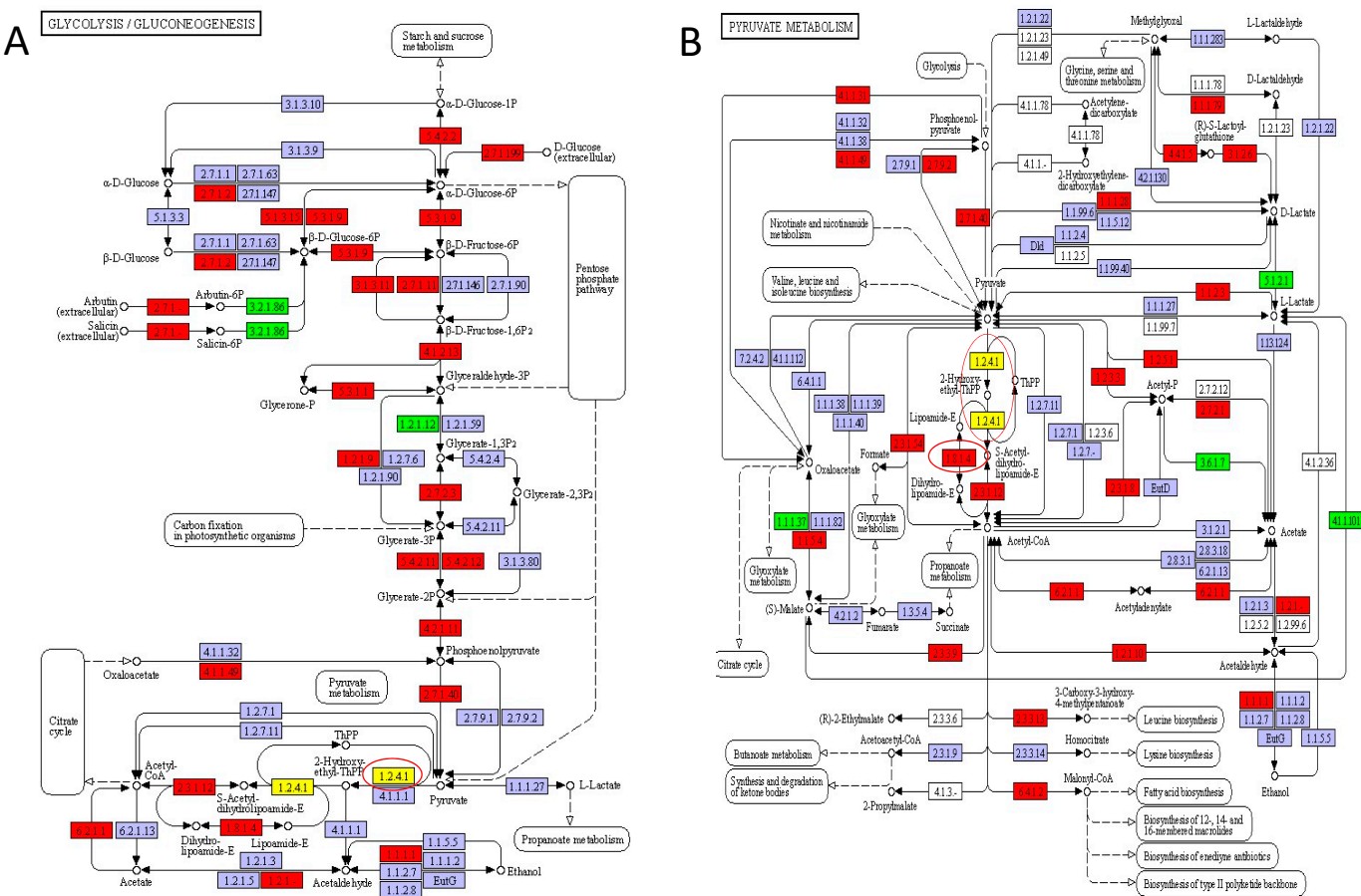

**Figure 7.** KEGG map of pathway: (**A**) glycolysis/gluconeogenesis (pathway 00010); (**B**) pyruvate metabolism (pathway 00620). Red circles represent the identified enzymes.

Besides glucose, the hexoses fructose and mannose were also metabolized sugar [55]. Carbohydrate transportation, particularly fructose and mannose metabolism, was identified in traditional Chinese sauerkraut [56]. As shown in Figure 8A, fructose and mannose metabolism were associated with the production of ascorbic, suggesting the metabolic pathway has a certain effect in the nutritional and flavor substances of DBSC. Moreover, amino contributes to the formation of organoleptic features of fermented products [50]. Other than taste characteristics of themselves, amino acids also metabolize organic acids, aldehydes, ketones, esters, and other products, which have a certain contribution to the color and aroma of fermentation product [21]. Figure 8B shows the glycine, gerine, and threonine metabolism pathway, which included threonine biosynthesis (aspartate => homoserine => threonine), serine biosynthesis (glycerate-3P => serine), ectoine biosynthesis (aspartate => ectoine), creatine pathway, cysteine biosynthesis (homocysteine + serine => cysteine), cetaine biosynthesis (choline => betaine), clycine cleavage system, cctoine-degradation (ectoine => aspartate) pathway modules, and other related metabolisms of alanine, aspartate, glutamate, cysteine, and methionine. Serine has a sweet taste and can be converted to acetic acid by deamidation reaction [57]. Amino acid can be converted to aldehydes, carboxylic acid, and alcohols, while aldehydes are subjected to further conversions into alcohols by dehydrogenation or into carboxylic acids by hydrogenation, which constitutes the formation of flavor [58]. Alanine can provide sweetness, and aminobutyric

acid is an amino acid derivative generated by the decarboxylation of glutamic acid [59]. Methionine and cysteine are responsible for the formation of methanethiol, sulfides, thioesters, and other sulfur-containing volatile compounds, which are mainly accountable to the aroma of Brassica vegetables [60]. Therefore, glycine, serine and threonine metabolism improve the flavor quality and nutritional value of DBSC.

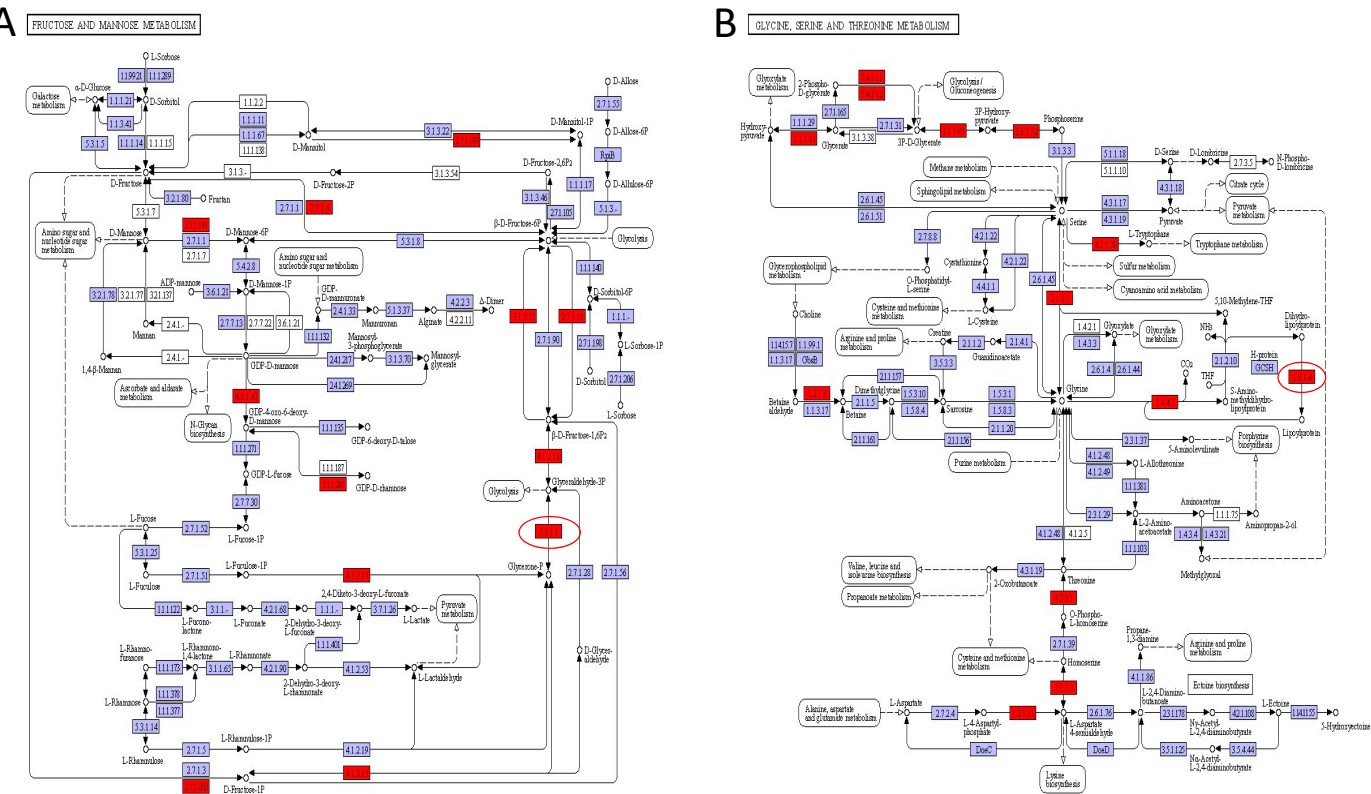

**Figure 8.** KEGG map of pathway: (**A**) Fructose and mannose metabolism (pathway 00051); (**B**) glycine, serine, and threonine metabolism (Pathway 00260). Red circles represent the identified enzymes.

## 5. Conclusions

In this study, metagenomic and metaproteomic approaches were jointly used to reveal the microbiological composition and protein profile and associated metabolic pathways of DBSC at different fermentation times. The bacterial communities were primarily dominated by firmicutes and proteobacteria in the phylum level and *Pseudomonas*, *Weissella*, *Pediococcus*, and *Leuconostoc* in the genus. In addition, functional metagenomic and metaproteomic profiling indicated that glycolysis/gluconeogenesis [path: ko00010], pyruvate metabolism [path: ko00620], fructose and mannose metabolism [path: Ko00051], and glycine, serine, and threonine metabolism [path: Ko00260] pathways were enriched. Moreover, critical proteins (oppA, rbsB, SOD$_2$, groEL, dps, fliC, tsf, fusA, atpD, metQ, pgi, tpiA, eno, alaS, bglA, tktA, gor, pdhD, aceE, gnd) related to metabolic pathways were identified.

This study provides an important insight into the microbial community structure and metabolic mechanism of DBSC, which provides a theoretical support for the DBSC fermentation industry.

**Supplementary Materials:** The following supporting information can be downloaded at: https://www.mdpi.com/article/10.3390/fermentation10040185/s1.

**Author Contributions:** Conceptualization: Y.Z., X.X. (Xiangxiu Xu), X.X. (Xiaowei Xiao), T.Z., M.L.; Methodology: Y.Z., Y.Y., N.G., L.Z., C.Z. and Z.W.; Software: Y.Z; Validation: Y.Z.; Data curation: Y.Z.; Writing—original draft: Y.Z.; Writing—review& editing: H.Y. (Haiqing Ye); Visualization: H.Y. (Haiqing Ye); Supervision: H.Y. (Haiyang Ya); Project administration: H.Y. (Haiqing Ye); Funding acquisition: H.Y. (Haiqing Ye).  All authors have read and agreed to the published version of the manuscript.

**Funding:** This work was supported by the National Natural Science Foundation of China [grant numbers 32372451].

**Institutional Review Board Statement:** Not applicable.

**Informed Consent Statement:** Not applicable.

**Data Availability Statement:** Data are contained within the article.

**Conflicts of Interest:** The authors declare no conflict of interest.

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
