# Peer review of "Metagenomic/Metaproteomic Investigation of the Microbiota in Dongbei Suaicai, a Traditional Fermented Chinese Cabbage"

_fermentation, doi:10.3390/fermentation10040185_

Round 1
Reviewer 1 Report
Comments and Suggestions for Authors
Manuscript deals on a study of the microbiome of Dongbei Suaicai a traditional fermented Chinese cabbage using metagenomic and metaproteomic approach. The objective of the manuscript is in accordance with the themes of the journal, and the results presented could help for a best understanding of this local product.
Comments:
1. Please increase the labels quality in each graph of the figures.
2. There is no clear conclusion to the work. The text refers to the objective of the work and a summary of the results, but the conclusions are missing.
3. A native English speaker must revise the manuscript. It is strongly recommended to use the English tools offered by the Editorial.
4. Additional specific comments are marked into the revised document using the Adobe acrobat tools.

A native English speaker must revise the manuscript. It is strongly recommended to use the English tools offered by the Editorial.
Author Response
Response to Reviewer 1 Comments
- Summary
Thank you very much for taking the time to review this manuscript. Please find the detailed responses below and the revisions highlighted/in track changes in the re-submitted files.
- Point-by-point response to Comments and Suggestions
Comments 1: [Please increase the label quality in each graph of the figures.]
Response 1: Thank you for pointing this out. We agree with this comment. Therefore, I have increased the quality of the labels in the graph of Figures 2,3,4, 5, and 6,7,8. In the revised manuscript, this change can be found in lines 172, 212,245,264, and 266, 410, 414.
Comments 2: [There is no clear conclusion to the work. The text refers to the objective of the work and a summary of the results, but the conclusions are missing.]
Response 2: Thank you for your opinions and constructive comments. We have revised the highlighted part in the Conclusion, hoping to meet your needs.
Comments 3: [A native English speaker must revise the manuscript. Additional specific comments are marked into the revised document using the Adobe Acrobat tools.]
Response 3: Thank you for your valuable comments on the Quality of the English Language, I have additional specific comments marked in the revised document using Adobe Acrobat Tools.
- Additional clarifications
Response 4: I will upload two files, one is a PDF with annotations and the modified Word. I hope these files will help you show me what I have revised and meet your requirements. Thank you again for all your comments.
Reviewer 2 Report
Comments and Suggestions for Authors
Manuscript fermentation-285918.
The manuscript studies the taxonomic and metabolic profiles of Dongbei Suaicai, a traditional fermented Chinese cabbage. It represents a piece of interesting information on the microorganisms involved in t fermentation at various levels, the diverse functional proteins found, and the diverse fermentation pathways to which they were related. The structure of the manuscript is correct; the language only requires minor corrections, and the expressions are clear and understandable. Some suggestions follow.
Supplementary material. Legend captions and the heading of columns should be included to facilitate the identification of contents.
In general, would it be necessary to write Day in capital letters?
In general, despite understanding that the great amount of information represents an important constraint, figure legends would be improved. Now is almost impossible reading them.
The names of some compounds (e.g., Glycine) are sometimes written in capital letters. Please revise.
Also, increases or decreases categories are expressed in percentages, but the differences are, in general, limited. Was no test applied to assess the significance of such differences?
Please change PH to pH
Figures. Remarks refer to figures. To remark on this linkage, the separation with the following text should be increased.
L-33. I guess that the identification number of the grant is only one
L-45. The paragraph is confusing
L-60. The paragraph sounds incomplete
L-75-77. The paragraph requires clarification.
L- 222. A wordy sentence. Obviously can be omitted.
L-405. Proportion. Capital letter?
L506. This acknowledgment should be in a separate section. Please revise the Journal Instructions.
Comments on the Quality of English Language
Requiere minor corrections.
Author Response
Response to Reviewer 1 Comments
- Summary
Thank you very much for taking the time to review this manuscript. Please find the detailed responses below and the revisions highlighted/in track changes in the re-submitted files.
- Point-by-point response to Comments and Suggestions
Comments 1: [Supplementary material. Legend captions and the heading of columns should be included to facilitate the identification of contents.]
Response 1: Thank you for your valuable comments. I have modified the legend and column headings in the supplementary materials to make it easier to identify the content. The original data of the experimental results are presented in the table to ensure the accuracy and reliability of the experimental results.
Comments 2: [In general, would it be necessary to write Day in capital letters?]
Response 2: Thank you for your opinions and constructive comments,after careful review, we have revised all "Day" to "day" throughout the article.
Comments 3: [In general, despite understanding that the great amount of information represents an important constraint, figure legends would be improved. Now is almost impossible reading them.]
Response 3: Thank you for pointing this out. We agree with this comment. Therefore, I have increased the quality of the labels in the graph of Figures 2,3,4, 5, and 6,7,8. In the revised manuscript, this change can be found in lines 172, 212,245,264, and 266, 410, 414.
Comments 4: [The names of some compounds (e.g., Glycine) are sometimes written in capital letters. Please revise.]
Response 4: Thank you for your valuable comments. This is really a mistake in our work. From line 466 to line 482, all relevant contents have been changed.
Comments 5: [Also, increases or decreases categories are expressed in percentages, but the differences are, in general, limited. Was no test applied to assess the significance of such differences?]
Response 5: Thank you for your valuable comments. We refer to other literature, and most of them don't use tests to evaluate the importance of this difference. Therefore, we decided not to adopt it either.
Comments 6: [Please change PH to pH]
Response 6: Thank you for your valuable comments. This is a mistake in our work. This content has been changed in the text.
Comments 7: Response to Comments on the Quality of English Language
We have polished and revised the English of the whole article. The modified part will be marked in red font. The following are some suggestions that you have highlighted, and we will reply to them one by one.
Point 1: L-33. I guess that the identification number of the grant is only one
Response 1: Thank your suggestion, the identification number of the grant is only one. I have removed the content of line 35.
Point 2: L-45. The paragraph is confusing
Response 2: Thank you for your very constructive comments. I have revised the sentences from line 48 to line 50 so that you can better understand the sentence in line 45.
Point 3: L-60. The paragraph sounds incomplete
Response 3: Your comments are of great help to us. I have put the paragraph where line 60 is located, the sentence has been modified and marked in red.
Point 4: L-75-77. The paragraph requires clarification.
Response 4: Thank you sincerely for your professional advice. The paragraphs where the sentences in lines 77 to 75 are located have been modified and marked in red.
Point 5: L- 222. A wordy sentence. Obviously can be omitted.
Response 5: Thank you for your advice, but the content of line 222 is a description of the result, and I don't think it can be omitted.
Point 6: L-405. Proportion. Capital letter?
Response 6: Thank you for your suggestion. This is really our mistake. It has been revised and marked in red font.
Point 7: L506. This acknowledgment should be in a separate section. Please revise the Journal Instructions.
Response 7: I quite agree with your comments. I have deleted line 506 from the manuscript.
- Additional clarifications
Response 4: I will upload two documents, one is the modification mode marked with red. One is a completely revised document. All revisions are marked in red font.